# The Optimization of a Natural Language Processing Approach for the Automatic Detection of Alzheimer’s Disease Using GPT Embeddings

**DOI:** 10.3390/brainsci14030211

**Published:** 2024-02-25

**Authors:** Benjamin S. Runde, Ajit Alapati, Nicolas G. Bazan

**Affiliations:** 1Science Engineering Research Center, The Potomac School, McLean, VA 22101, USA; 2Neuroscience Center of Excellence, School of Medicine, New Orleans, LA 70112, USA; ajitalapati@gmail.com

**Keywords:** dementia, NLP, GPT, embeddings

## Abstract

The development of noninvasive and cost-effective methods of detecting Alzheimer’s disease (AD) is essential for its early prevention and mitigation. We optimize the detection of AD using natural language processing (NLP) of spontaneous speech through the use of audio enhancement techniques and novel transcription methodologies. Specifically, we utilized Boll Spectral Subtraction to improve audio fidelity and created transcriptions using state-of-the-art AI services—locally-based Wav2Vec and Whisper, alongside cloud-based IBM Cloud and Rev AI—evaluating their performance against traditional manual transcription methods. Support Vector Machine (SVM) classifiers were then trained and tested using GPT-based embeddings of transcriptions. Our findings revealed that AI-based transcriptions largely outperformed traditional manual ones, with Wav2Vec (enhanced audio) achieving the best accuracy and F-1 score (0.99 for both metrics) for locally-based systems and Rev AI (standard audio) performing the best for cloud-based systems (0.96 for both metrics). Furthermore, this study revealed the detrimental effects of interviewer speech on model performance in addition to the minimal effect of audio enhancement. Based on our findings, current AI transcription and NLP technologies are highly effective at accurately detecting AD with available data but struggle to classify probable AD and mild cognitive impairment (MCI), a prodromal stage of AD, due to a lack of training data, laying the groundwork for the future implementation of an automatic AD detection system.

## 1. Introduction

Alzheimer’s disease (AD) is an incurable neurological disorder that causes the degeneration of neurons in the brain, progressing first from dementia to the eventual inability of the brain to conduct basic bodily functions. AD incidence is increasing and is projected to rise to 13.8 million Americans by 2060 [1]. AD is the most common form of dementia, making up an estimated 60% to 80% of global cases of dementia. With the world population aging, people over the age of 65 are expected to increase by 50% halfway through the century, and the social and economic impact of AD is expected to grow rapidly. Current data suggest that 68% of this growing impact is expected to occur in low- and middle-income countries. Age and heredity are the two key risk factors for the onset of AD. However, the understanding of AD etiopathogenesis remains an enigma. There is an unmet need to uncover the underlying mechanism(s) of the onset and progression and to identify biomarkers associated with onset to be able to screen vulnerable populations, diagnose at-risk patients early, and monitor progression and response to therapeutics [2].

Being a progressive disease, AD manifests initially with preclinical AD through subjective cognitive impairment (not all cases transition to AD), then mild cognitive impairment (MCI), and finally dementia (which continually worsens over time), making it paramount that the disease be detected as early as possible in order to slow its progression and impact [1]. Currently, the diagnosis of AD using conventional clinical methods requires a specialty clinic, which can be invasive, expensive, and time-consuming. Additionally, these methods are often inaccurate and not cost-effective, particularly in identifying the early stages of the disease. Furthermore, nonspecialist clinicians often struggle to accurately identify early AD and MCI. As a result, there is a growing demand for noninvasive and/or cost-effective tools that can identify individuals in the preclinical or early clinical stages of AD, allowing for early interventions that could improve lifestyle and for the evolution of pharmacological treatments. This is particularly important for lower-income individuals who may have fewer resources to cope with AD; therefore, a more effective, accurate, and cost-effective way of detecting early AD is necessary [3].

As the stages of AD progress, aphasia (the inability to understand or formulate language) and dysgraphia (the inability to write), some of AD’s most common symptoms, become worse, being marked by a predictable set of changes. Firstly, language and speech are impaired by the inability to find certain words, most commonly those pertaining to items or people the patient interacts with often, causing an increase in the use of pauses and filler words. In later stages, these symptoms are exacerbated, and the patient’s verbal acuteness and fluency are significantly impaired [4]. While some studies have shown that not all facets of speech and language change drastically in the first stages of the disease, the linguistic quality and complexity of the content of patients’ speech does, making it possible for artificial intelligence (AI) to conduct natural language processing (NLP) tasks for the automatic detection of AD (ADAD), based partially or entirely on the patient’s language [5,6].

NLP is a cross-disciplinary technique that aims to enable AI, specifically through large language models (LLMs), to understand and process text, enabling it to convey meaning to other models that can create summaries, responses, or, in this case, classify text. Thanks to the massive advances in LLMs and AI as a whole, in recent years, NLP methods have improved drastically, enabling models to understand deeper and more complex semantic features [7]. To perform NLP, most models use word embeddings, which are N-dimensional vector representations of words (Figure 1). Embeddings allow neural networks (NNs) and other machine learning classifiers (MLCs) to process language through semantic meaning, unlike other techniques that focus on the frequency of specific words, among other aspects [8]. One of the most advanced LLMs is OpenAI’s Generative Pre-trained Transformer 3 (GPT-3), which is known for its use in ChatGPT. Based on the GPT-3 architecture, OpenAI offers a set of highly advanced, cost-effective embedding models [9]. First-generation versions of these models have shown promising results when it comes to the NLP-based automatic detection of AD [10].

Past research into the automatic detection of AD using speech has focused on using either acoustic features or NLP techniques [10,11]. While acoustic feature-based models have been shown to perform effectively, achieving accuracies of 63.6% in Chlasta and Wolk’s work [12] using a convolutional neural network (CNN) and 65.6% in Balugopalan and Novikova’s work [13] using a support vector machine (SVM) classifier, Balugopalan and Novikova showed that a word embedding or combination approach was more effective. They performed better in nearly all metrics using several machine learning classifiers, achieving an accuracy of 66.9% for embeddings and 69.2% for the combination using SVM. Cruz et al. [14] used NLP techniques, specifically sentence embeddings, using Siamese BERT-Networks (SBERT) to create embeddings and test the effectiveness of several types of ML classifiers. They found that SVMs and neural networks (NNs) were the most effective, achieving accuracies and F-1 scores (the harmonic mean of precision and recall) of 0.77 and 0.80 (SVM) and 0.78 and 0.76 (NN), respectively.

Agbavor and Liang built upon the research of both Balugopalan and Novikov and Cruz et al. [14] Using audio files from the ADReSSo dataset, they extracted acoustic features, and they converted audio to text automatically using a transcription program, extracting embeddings using OpenAI first-generation embedding models. Using these acoustic features and embeddings, they trained multiple models using different combinations of NLP methods and ML classifiers. When comparing models, they found that the most effective model produced used only word embeddings, and it was classified using an SVM. This model was able to achieve an accuracy of 0.803 and 0.829 for accuracy and F-1 [10]. These results demonstrate that the integration of SVM classifiers with advanced word embeddings constitutes one of the most efficacious approaches in the scientific domain for automatic Alzheimer’s disease detection.

This study aims to build on past research and optimize an NLP-based automatic AD detection system, increasing its performance. By optimizing the methods required to implement one of these systems, we hope to characterize the full potential of this technology in its current form while also identifying areas of improvement necessary to assist in the creation of a real-world application. Specifically, using audio files from the Pitt Corpus of the Dementia Bank Database, we aim to optimize the transcription process to increase the quality of the GPT word embeddings and the subsequent classification models that they train [15,16]. We will exclusively use GPT word embeddings and SVM classifiers to test the effectiveness of these techniques due to their proven efficacy in previous studies. To optimize these methodologies, we seek to evaluate the performances of several AI-based audio transcription systems, using cloud- and locally-based transcription services, in addition to an audio enhancement system to aid in automatic transcription. We also aim to compare the performance of manual transcripts to those made with AI and seek to understand the impact of including interviewers in recordings. Using these various methodologies, we will characterize their performances in various classification tasks utilizing different diagnosis types.

## 2. Methodology

The overall approach of this study can be explained as follows, and a visual overview of the process can be found in Figure 2.

### 2.1. Database Information

For the study, we used the Pitt Corpus, which can be found in the Dementia Bank Database [15]. Dementia Bank is a database that is part of the Talk Bank project, which collects and makes available several different types of multimedia files that relate and can contribute to the study of language and communication of dementia [16]. The Pitt Corpus, which is derived from Becker et al. [15], was gathered as part of a larger project to study dementia at the University of Pittsburgh School of Medicine. According to the data sheet available with the Pitt Corpus, the dataset included 244 samples of Probable AD, 87 samples of Possible AD, 16 samples of Vascular Dementia, 6 samples of other dementias, 12 samples of people who had cognitive problems yet lacked a diagnosis, 23 samples of MCI, and 121 samples of a Control group.

For every individual interview (sample), an original audio file, an enhanced audio file, and a written transcript in CHAT file format of the patient describing the Cookie Theft image (Figure 3) were included as well [15]. The Cookie Theft image is an image included in a sub-test of the Boston Diagnostic Aphasia Examination that has risen to prominence thanks to its potential to reveal a wide range of cognitive and linguistic skills and deficits [18]. For this sub-test, patients are shown a drawing of a mother cleaning dishes next to the sink. They are instructed to tell the interviewer all that they see going on in the picture. The Cookie Theft picture contains a wide range of describable features, including people, objects, and actions [19].

### 2.2. Organizing Database Data

Upon accessing the files, we immediately noticed a discrepancy between the quantities of samples listed and those actually available. This meant that it would be impossible to sort through the included files using the available data sheet. Instead, we opted to write a program using the Python (3.11.4) programming language that separated all of the original (or standard quality) and enhanced audio files as well as the manual transcripts by diagnosis type using the diagnosis information available in the CHAT file format of the transcripts. Once both types of audio files and the CHAT transcripts were organized by diagnosis type, we re-counted the total for each diagnosis type; we found 234 samples of Probable AD, 21 samples of Possible AD, 42 samples of MCI, 3 samples of MCI with only memory problems, 5 samples of Vascular Dementia, 1 sample with another diagnosis, and 242 samples of Control. Using this information, we removed the MCI with memory problems only, vascular dementia, and other diagnosis groups, as they lacked enough data to train and test a model.

### 2.3. Audio Enhancement

Included in the Pitt Corpus were the original and enhanced versions of each interview’s audio file [15]. Audio files were enhanced by removing background frequencies using an implementation of Boll Spectral Subtraction available for Mathworks MATLAB (R2020a) program [17,20]. Boll Spectral Subtraction works by assuming background frequencies and subtracting them from the original audio file. Spectral Subtraction offers a computationally efficient, consistent, and effective way of removing consistent background frequencies, but it is not able to remove inconsistent and random audio artifacts [21]. This implementation of Boll Spectral Subtraction uses the first 0.25 s of audio, which is presumed by the program to be representative of background frequencies, and estimates the average background noise frequency using spectral averaging. Using this estimated frequency or range of frequencies, it subtracts them from the original audio file. Following this, a secondary residual noise reduction is done to enhance the quality of the audio files [17].

### 2.4. Manual Transcript Processing

Manual transcripts included in the Pitt Corpus data are complete documentation of the interview, including the interviewer’s questions and the patient’s responses. For example, the included transcript for interview ID 002-1 starts with the interviewer asking, “What do you see going on in that picture?” and the patient responds with, “Oh, I see the sink is running” [15]. Since the goal of the study is to optimize an NLP approach to the automatic detection of AD, removing the healthy, unaffected interviewer would remove any erroneous data that could hurt the performance of the models [22]. While it would be nearly impossible to differentiate between the interviewer and the participant in an automatic transcript, the CHAT format of the included manual transcripts indicates the speaker for every line of text. Using this, we wrote a program in the Python programming language that created a complete, unchanged transcript and a version with the interviewer removed. These new transcripts were exported in Excel format and only included text characters, removing any special characters included in the transcripts for CHAT file formatting conventions.

### 2.5. Automatic Audio Transcription

The original, or standard quality, and enhanced audio files were converted to text transcripts using four separate automatic speech recognition (ASR) programs (Figure 4).

The first program that we used was a trained Wav2Vec model. This model showed promising results in Agbavor and Liang [10]. The specific model that was used was the larger, most advanced model, facebook/wav2vec2-large-960h, which was trained and fine-tuned for transcription accuracy on 960 h of Librispeech on 16 kHz sampled speech audio [23]. This model can be found on the Hugging Face platform [24]. Audio files were transformed into waveforms using the Librosa library for Python [25]. Then, using the Wav2Vec2Tokenizer (4.3.0), waveforms were parsed into smaller, more accessible, and computationally efficient sections. These sections were then converted into text using the Wav2Vec2ForCTC sub-model, which inherits and learns from the selected pre-trained model. Once all transcripts were created for both enhanced and standard audio, they were exported in Excel format.

The second model used for generating automatic transcriptions using ASR was Rev AI. This method, proposed by the Talk Bank project, attempts to streamline an efficient and user-friendly way of creating high-quality automatic transcriptions [26]. The user interface is created through a program called Docker, which creates an access portal on one’s own device to upload files [27]. Then, through the Docker portal, one uploads their Rev AI API key, allowing the interface to send the files to the Rev AI service, an industry-leading ASR program [28,29]. Once the files are converted to text, they are immediately downloaded to one’s computer in the CHAT file format. The Rev AI CHAT transcripts were then converted into Excel format.

The third model we used was OpenAI’s Whisper program. Whisper is an open-source, locally run ASR model that is designed to excel in a zero-shot learning environment. This means it is designed to work effectively without requiring a program to be prepared by training it with a downstream task through an approach such as fine-tuning, where one gives the pre-trained model a secondary dataset (in this case, a set of audio files and their correct transcripts) so that it can adjust to its task. Whisper was trained using 680,000 h of multilingual and multitasking supervised data from the internet, allowing it to succeed in standard benchmarks in multiple languages [30]. Using a program written in the Python programming language, audio files were processed through the Whisper model, and the subsequent transcripts were exported in Excel format.

The final model we used for the transcription of the standard and enhanced audio files was the IBM Cloud-based Watson Speech-to-Text (STT) service. An API key was created using IBM Cloud’s web interface. Using this API key, as well as Librosa, to tokenize and partition audio files, we created a program in Python that accessed the Watson Speech-to-Text base model through the cloud [25,31]. Once transcripts were created, they were exported in Excel file format.

Of the ASR services used in this study, two were cloud-based—IBM Cloud Watson STT and Rev AI—and two were open-source and locally based—Wav2Vec and Whisper [23,28,30,31]. The cloud services are thought to be more advanced but require payment, using a pay-as-you-go model, as computations were performed remotely through each company’s own servers and dedicated hardware. IBM Cloud Watson STT and Rev AI both used an affordable pricing scheme of USD 0.02 per minute of audio transcribed by each service [28,31]. OpenAI Whisper and Hugging Face Wav2Vec transcribed files locally using the computer’s own hardware and were free to use. For each type of ASR service, whether cloud or local, one service was selected for its use or proposed in past ADAD research—namely, Rev AI for the cloud base set and Wav2Vec for the local set [10,26]—and one was selected for its industry-leading performance—namely, IBM for cloud-based and Whisper for local [30,31].

### 2.6. Aggregation of Transcripts

Once the manual transcripts were processed and the audio files were transcribed, we combined and organized all of the new transcripts based on each interview. For each interview, there were 10 transcripts that could be used to train separate models to compare transcript methodology performances. The final transcript types that we combined and used were as follows: Unchanged Manual Transcript, Manual Transcript Interviewer Removed (also known as participant-only), Wav2Vec Standard, Wave2Vec Enhanced, Rev AI standard, Rev AI enhanced, Whisper Standard, Whisper Enhanced, IBM Standard, and IBM Enhanced. These transcripts were all combined in an Excel spreadsheet, where each row included interview information and 10 subsequent transcriptions using each methodology. Interviews that were unable to be transcribed through one of the methodologies were dropped from the data. In total, 18 interviews were removed: 9 from control, 7 from AD, 2 from MCI, and 0 from Possible AD. The final sizes of each diagnosis group in this study were 233 samples of Control, 227 samples of Probable AD, 40 samples of MCI, and 21 samples of Possible AD (Table 1).

### 2.7. Creation of Embeddings

Embeddings were created using the OpenAI second-generation embedding model, called text-embedding-ada-002. An interpretation of word embeddings can be seen in Figure 1. First proposed by Agbavor and Liang [10], the first-generation OpenAI embedding models showed promising results, contributing to an approach that achieved an accuracy of 80.3%. Using the Python Pandas library, a data analysis package for Python (3.11.4), the combined transcripts were loaded as a data frame [32]. Using this data frame and an OpenAI API key, we created an embedding program for all the transcripts using the second-generation embedding model through API requests to OpenAI’s servers [9]. Pricing for the OpenAI second-generation embedding model is USD 0.0004 per 1000 tokens (which is slightly less than a word) or around 3000 pages per USD 1, much cheaper than the various first-generation models, which had worse performance and ranged from 6 to 300 pages per USD 1 [33].

### 2.8. SMOTE

Once embeddings were created, we applied the Synthetic Minority Over-sampling Technique (SMOTE) to balance out the datasets. Balanced datasets are essential for machine learning classifier performance [34]. SMOTE can be accessed through the imbalanced-learn library for Python [35]. SMOTE is an algorithm that performs data augmentation and balancing by creating synthetic data based on the original minority data points. SMOTE works by selecting random minority data points, estimating their Euclidean distance from their k-nearest neighbors, multiplying the distance between the parent point and each k-nearest neighbor by a random number between 1 and 0, and then adding up those values to create a vector that is applied to the parent data point to create the synthetic one [34]. However, since text-embedding-ada-002 generates embeddings with a dimensional size of 1536, we acknowledge the potential limitations of using SMOTE, as it is most effective within feature sizes below 1000, introducing possible distortions and bias into the data [33,36]. Future and more advanced embedding models that produce embeddings with fewer but higher quality and more useful dimensions or more advanced data augmentation techniques tailored for high-dimensional data would be more effective. Simply, SMOTE estimates the general area of the minority samples and creates synthetic samples in that general area to balance out the datasets. SMOTE was applied to the MCI and Possible AD diagnosis types, increasing their sample sizes from 40 and 21 to 100 each (Figure 5). The final size of each diagnosis type, including synthetic data, is 233 samples of Control (unchanged), 227 samples of Probable AD (unchanged), 100 samples of MCI, and 100 samples of Possible AD.

### 2.9. Data Subgroups for Classifier Models

Since SMOTE is not a perfect technique for data augmentation, as it still relies on past data to create synthetic data, and since the number of dimensions of our embeddings exceeds SMOTE’s most effective dimensional range, some degree of bias will be introduced into models using data augmented by SMOTE. Therefore, for the proposed comparisons that this study is trying to achieve, we created several models for each transcription methodology using different combinations of diagnosis types (Table 2). The first data subgroup that we used for model training included all the Control and AD samples (approximately 230 each). This set of data gave us the most unbiased results, as it lacked any synthetic data and used all the data samples available for those two subgroups. The second is a subgroup that only used the downsized sample sizes for Control and AD (100 each). This subgroup lacks any bias from synthetic data but does not use all the data available (for Control and AD) so that it can be used for comparisons with other studies that have similar sample sizes. The third is a subgroup that only uses the downsized sample sizes for Control and AD and uses the synthetically up-scaled MCI sample size (100 each). These data will have some bias, as the MCI data type has been augmented with SMOTE and not all the samples of AD and Control will be used, as the sample size for each class needs to be equal. The final subgroup used 100 samples of all the data types: Control, AD, MCI, and Possible AD. This model will have the most bias since two of its classes have been augmented using SMOTE.

For the last two subgroups (AD, MCI, and Control (100×) and AD, MCI, Possible AD, and Control (100×)), which use synthetic data due to the lack of data available, both the train–test split and the 10-fold CV were run after data augmentation, since there were not enough samples to split data first and then augment the data with each separated group. This circumstance increases the bias produced by data augmentation, since features extracted from an original sample could appear in a synthetic sample in the other class.

### 2.10. SVM Training and Testing

For diagnosis classifications, this study used a support vector classifier (SVC). A visual interpretation of an SVC can be seen in Figure 6. In Agbavor and Liang [10], SVCs were shown to have the best classification performance when compared to Random Forest (RF) and Logistic Regression (LR) classifiers for the binary classification of AD and Control. Building upon this research, we have chosen to train various SVCs using all four subgroups for every transcription group/methodology. SVCs and SVMs can be accessed using the SciKit-Learn platform and Python library [37]. The NumPy and Pandas Python libraries were imported and used to format and process data/results [32,38], and the Matplotlib Python library was used to export model performances in a graphical format [39].

The first model that was trained for every data subgroup used an 80/20 train–test split. Commencing with data preprocessing, the dataset was divided into distinct components, namely an 80% training set and a 20% testing set. To characterize the trained models’ full capabilities and potentials, we used the capabilities of the GridSearchCV object, which systematically traversed an array of parameter combinations (regularization parameter C, kernel selection, polynomial degree (where relevant), and the kernel coefficient gamma) through cross-validation (CV), finding the most effective settings for each model. Upon successful completion of the tuning process, the highest-performing model was automatically selected, and it was subsequently retrained using the determined optimal hyperparameters. For hyperparameter tuning, only the training data were used. Using this optimally tuned SVM classifier, the model performance was quantified using the unseen test data. All models (for each transcript methodology) used the transcripts of the same interviews for their own training and testing samples to allow for a more accurate and direct comparison.

A second SVM classifier was created to test model generalizability using a 10-fold cross-validation technique. We executed an 80/10/10 train–validation–test split to rigorously evaluate the performance of a support vector machine (SVM) classifier with a linear kernel. In this code, we performed k-fold cross-validation, where k is set to 10, to evaluate the performance of a support vector machine (SVM) classifier with a linear kernel. The dataset was initially split into 10 approximately equal and stratified subsets. Each of these subsets, referred to as “folds”, played a distinct role in the cross-validation process. During each iteration of the loop, one fold served as the validation set, while the remaining nine folds were used for training a linear SVM model. The “random_state” was set for each fold to ensure reproducibility and uniqueness. With each trained model, we then made predictions on the validation set and assessed its performance. The results of each fold, encompassing all the performance metrics, were collected in separate lists, allowing for the evaluation of the SVM model’s ability to generalize effectively across different subsets of the data.

## 3. Results

### 3.1. Performance Metrics

The main performance metrics used by this study are accuracy, precision, recall, and F-1 score. These metrics are commonly used and are the de facto standard for quantifying machine learning classification performance. These metrics are built on the True/False Positive (TP) (FP) and True/False Negative (TN) (FN) values of each model. Accuracy quantifies the overall percentage of samples that were correctly classified. Precision is a metric that reveals what percentage of the samples marked true are, in fact, true. Recall is a metric that reveals what percentage of true samples were marked as TP. F-1 score combines precision and recall, representing their harmonic mean [40].
(1)$$Precision=\frac{TP}{TP+TN}$$
(2)$$Recall=\frac{TP}{FN+TP}$$
(3)$$Accuracy=\frac{TP+TN}{FP+FN+TP+TN}$$
(4)$$F1=\frac{2\cdot Precision\cdot Recall}{Precision+Recall}$$

### 3.2. Results for 80/20 Train–Test Split

The complete results of all models (for all data subgroups within each transcription methodology) using the 80/20 train–test technique can be found in Table 3, Table 4, Table 5 and Table 6. The train–test split test reveals the performance of a trained and optimized model on an untrained test set. This simulates the potential of the model to perform on a real-life test set when it is optimized with more real-life data; in other words, the performance the model could achieve if more data were collected or its peak possible performance. It does this by tuning hyperparameters by training and testing dozens of models on the original training set to simulate the enhancement of the model. These results are measured using accuracy, precision, recall, and F-1 score. Table 3 through Table 6 are divided up firstly by ASR program, then by type of audio or manual transcription, and then by each data subgroup. This test can be used to make the most direct and accurate comparisons possible, as each methodology will have the same interviews for their training and testing data. Comparisons between models are only applicable within each data group, since each group uses a different dataset size and complexity.

For the AD and Control (230×) subgroup, accuracy ranged from 0.79 to 0.99, and F-1 scores ranged from 0.78 to 0.99 (Table 3). The data for the AD and Control (230×) can be found in Table 1, which shows all models’ performance using the train–test split test, as well as Table 3, which only includes data from the AD and Control (230×) models using the train–test split. The best-performing model was the Wav2Vec Enhanced model. It achieved an accuracy of 0.99 and an F-1 Score of 0.99. The second best model was the Rev AI Standard model. This model achieved an accuracy of 0.96 and an F-1 score of 0.96. The worst-performing model was Rev AI Enhanced. This model only managed to achieve an accuracy of 0.79 and an F-1 score of 0.78. The second worst-performing model was the model using the Manual Transcripts Unchanged, which achieved an accuracy and F-1 score of 0.87.

For the AD and Control (100×) subgroup, performance overall improved, with accuracy and F-1 scores ranging from 0.84 to 1.00 (Table 4). The best-performing models were Rev AI Enhanced and Wav2vec Enhanced, which both scored 1.00 for accuracy and F-1. The second-best models were IBM Cloud Standard and Rev AI Standard, which both achieved an accuracy and F-1 score of 0.98. Manual Transcript Participant Only and IBM Cloud Enhanced were tied for second worst, both scoring 0.89 for accuracy and 0.88 for F-1 score. The worst-performing model for the AD and Control (100×) subgroup was Wav2Vec Standard, which scored 0.84 for accuracy and 0.84 for F-1 score.

The performance continued to increase for the third data subgroup—AD, MCI, and Control (100×)—as seen in Table 5, ranging from an accuracy and F-1 score of 0.98 and 0.98 to 0.90 and 0.90, a smaller range than the previous models. The most effective model was the Manual Transcript Participant Only, which scored 0.98 for accuracy and 0.98 for F-1. The next best model was Manual Transcript Unchanged, with an accuracy of 0.97 and 0.97 F-1. For the third best, there was a tie between IBM Cloud Standard and Wav2Vec Enhanced, which both scored 0.95 for accuracy and F-1 score. The second worst model was Rev AI, with an accuracy of 0.91 and an F-1 score of 0.90. This was followed by IBM Cloud Enhanced, the worst model, which scored 0.90 for accuracy and F-1 score.

The overall performance for the last data subgroup—AD, MCI, Possible AD, and Control (100×)—decreased from the last group, but the range stayed consistent, ranging in accuracy and F-1 score from 0.96 and 0.96 to 0.88 and 0.87 (Table 6). The best two models for this subgroup both used the IBM Cloud ASR program. The best was IBM Cloud Enhanced, which scored 0.96 for accuracy and 0.96 for F-1 score. IBM Cloud Standard was second, scoring 0.95 for accuracy and 0.95 for F-1 score. The second worst model was Wav2Vec Standard, which scored 0.89 for accuracy and 0.88 for F-1. The worst was Rev AI Enhanced, which scored 0.88 for accuracy and 0.87 for F-1 score.

### 3.3. Complete Results for 10-Fold Cross-Validation

The 10-fold cross-validation was used to assess and evaluate the ability of each machine learning model to generalize using the average results of 10 machine learning models using a different test set each time. This model gives a more realistic result of how a model will perform in its current form on a real-life test set, while the train–test split shows the model’s potential performance. Because of the nature of cross-validation, the random groups or folds it splits the data into make it less accurate at making direct and precise comparisons of models that use the same data.

The overall results for the 10-fold cross-validation tests were lower across the board than the train–test split, as can be seen in Table 3, Table 4, Table 5 and Table 6. For the AD and Control (230×) subgroup, the best performing model was Manual Transcript Participant Only, which scored 0.84 for accuracy and 0.84 for F-1 score (Table 3). The second best was Manual Transcript Unchanged, which scored 0.82 for both accuracy and F-1 score. The worst model was IBM Cloud Enhanced, which only scored 0.72 for accuracy and F-1 score. The overall scores were lower, and the range, which spanned from 0.84 to 0.72 for both accuracy and F-1 score, was smaller than the train–test split test.

For the AD and Control (100×) subgroup, the best performing model was Manual Transcript Participant Only, which scored 0.80 for accuracy and 0.79 for F-1 score (Table 4). The second best was Whisper Standard, which scored 0.79 for both accuracy and F-1 score. The worst model was Rev AI standard, which scored 0.63 for accuracy and 0.60 for F-1 score. The scores for this subgroup ranged from 0.80 to 0.63 for accuracy and 0.79 to 0.60 for F-1. For the third subgroup—AD, MCI, and Control (100×)—the overall performance continued to decrease (Table 5). The best model was Whisper Enhanced, which scored 0.56 for accuracy and 0.55 for F-1. The worst model was Rev AI Standard, which scored 0.45 for accuracy and 0.43 for F-1 score. The size range decreased, only spanning from 0.56 to 0.45 for accuracy and 0.55 to 0.43 for F-1 score. The fourth subgroup—AD, MCI, Possible AD, and Control (100×)—performed the worst in this test, having scores ranging from 0.58 to 0.44 for accuracy and from 0.56 to 0.41 for F-1 score (Table 6). The best model was the Wav2Vec Standard, which had a score of 0.58 for accuracy and 0.56 for F-1 score. The worst was Rev AI Standard, which scored 0.44 for accuracy and 0.41 for F-1 score.

## 4. Discussion

### 4.1. Data Used for Direct Comparisons and Observations of Transcription Methodologies

The data used for making direct comparisons between transcription methods are the AD and Control (230×) subgroups. AD and Control (230×) are also used to posit the most effective model as a whole created by this study. This group has the lowest amount of bias, as it does not include any synthetic data and uses all the data available to it. Furthermore, for comparisons between models, we use the data from the train–test SVM classifier, as all models (within the same subgroup) used samples from the same exact interview for their respective training and testing groups, which is not the case for the cross-validation test.

AD and Control (100×) are used to make overall comparisons to other studies that have predominantly used smaller databases of a similar size to this data group, such as the ADReSSo challenge and dataset, which has a size of 237 samples, around 120 samples per group. ADReSSo is a recurring competition that aims to create the best model for detecting and differentiating between AD and Control diagnoses using any audio-based method [41]. This subgroup lacks any bias from synthetic data, but since it does not use all of the data available to it, it cannot find the most accurate results possible for each methodology and thus is not used for direct comparisons between methodologies.

The last two data subgroups are used to analyze the preliminary possibility of detecting MCI and Possible AD, as these subgroups include some degree of bias from synthetic data.

### 4.2. AD and Control (100×) Subgroup Results and Comparisons with Previous Studies

The results of the AD and Control (100×) subgroup are very promising. As stated earlier, four models achieved perfect or near-perfect results: Rev AI Enhanced and Wav2vec Enhanced, which performed perfectly (accuracy and F-1 of 1.00), as well as IBM Cloud Standard and Rev AI Standard, which were near-perfect (accuracy and F-1 of 0.98). Five of the remaining models achieved scores in the low 0.90 s and high 0.80 s, which are still extremely impressive. The Wav2Vec Standard scored 0.84 for accuracy and F-1 score, which was still quite good despite being the worst-performing model. When comparing the train–test split scores to the cross-validation results, they were overall much lower, only ranging from 0.80 to 0.63 for accuracy and 0.79 to 0.60 for F-1 score. This discrepancy is most extreme for some of the best-performing models in the train–test split test, which performed near the bottom for cross-validation. This is the case for the Rev AI Standard and Enhanced, which only scored 0.63 and 0.65 for accuracy and 0.60 and 0.64 for F-1 score, and not for Wav2Vec Enhanced and IBM Cloud Standard.

While this discrepancy between the performance of the train–test split and cross-validation in the Rev AI models is a possible indicator of overfitting, usually caused by a data leakage or a dataset that is too small (which this dataset is at risk of), the results of the Wav2Vec Standard model give the other results of this data subgroup credence for comparisons with other studies [42]. This is because, when examining the results of Agbavor and Liang [10], who used a methodology nearly identical to the Wav2Vec Standard, which included training on the ADReSSo dataset (120 for both AD and Control), using the standard audio of the study, and using Wav2Vec transcriptions, which were turned into embeddings using the GPT first-gen models, their performance is very similar. While we achieved 0.84 for accuracy and F-1 (train–test) for the Wav2Vec Standard methodology, they scored 0.803 for accuracy and 0.829 for F-1 using an SVC [10]. Therefore, while some of the models are suffering from overfitting due to their poor generalizability when compared to train–test data, the similar performances of the Wav2Vec Standard methodology show that the rest of the transcript methodologies are much more effective overall than the previously used Wav2Vec Standard transcription methodology. This large improvement in performance indicates that, through the optimization of transcriptions, the performance of embedding-based AD detection programs can be improved dramatically.

### 4.3. Interpretation of the AD and Control (230×) Subgroup

Overall, the performance of the models using the AD and Control (230×) subgroup remained excellent. The best model by far was Wav2Vec Enhanced, which achieved an accuracy and F-1 of 0.99. Besides both Manual Transcript methodologies, Rev AI Enhanced, and Wav2Vec Standard method, the remaining five methodologies (IBM Cloud, OpenAI Whisper, and Rev AI Standard) had excellent performances, all achieving F-1 and accuracies between 0.91 and 0.96, which still outperform almost all other automated AD detection systems. The performance of both Manual Transcripts and Wav2Vec Standard was still quite good, scoring just below 0.90 in the upper 0.80s. The only poorly performing model was Rev AI Enhanced, which only was able to score an accuracy of 0.79 and an F-1 score of 0.78.

When compared to the results of the AD and Control (100×) subgroup, the performance of the AD and Control (230×) subgroup was much more consistent, which is to be expected when a larger sample size is used. The best-performing model was still Wav2Vec Enhanced, whose accuracy and precision only decreased by 0.01, to 0.99, when using more data samples. The other models that performed extremely well using the smaller dataset, Rev AI Enhanced/Standard and IBM Cloud Standard, had their train–test performances decrease and their CV performances increase. While Rev AI standard and IBM Cloud Standard were still the second and third best models using the train–test Split and the larger dataset, Rev AI Enhanced became the worst, having a similar train–test split performance as its CV scores (0.77 for accuracy and 0.76 for F-1). When we compare the train–test Split results to the CV results, the gap is smaller with the larger dataset (AD and Control (230×)) than with the smaller one (AD and Control (130×)). Similarly, the best-performing models for the larger dataset using the train–test split test (Wav2Vec Enhanced, Rev AI Standard, and IBM Cloud Standard) were not the worst models when it came to the CV, all scoring near the middle of the pack. Since the gap between the train–test Split and CV results decreased and the overall performances between both tests became more consistent, the larger database clearly helped mitigate the overfitting experienced by the models using the AD and Control (100×) data subgroup.

### 4.4. Negative Impact of Interviewer on Model Performance

When comparing both of the Manual Transcript methodologies, one can observe that there is a minor difference in performance. While the Manual Transcripts Unchanged model scored 0.87 for both accuracy and F-1, the Manual Transcripts Participant Only model scored 0.89 and 0.88 for accuracy and F-1, respectively. This improvement in performance indicates that it could be advantageous in the long run to remove interviewers from audio transcripts. This could be done in three ways, either by instructing the interviewer to begin the recording after the instructions, having the interviewer say a start and stop phrase between questions (so that their words could be removed), or through some sort of AI implementation (through voice recognition technology). Since more data are needed to more thoroughly test some of the methods proposed by this study and others so that a real-world application can be made, these suggestions should be taken into consideration when collecting data for a new database.

### 4.5. AI Transcription Models Outperforming Manual Transcripts

Interestingly, almost all of the AI-based ASR methodologies outperformed the pre-existing manual transcripts, despite some transcripts not having the same quality as the manual transcripts. For example, one phrase was manually transcribed as “the scene is in the in the kitchen, the mother is wiping dishes”, while Wav2Vec Standard transcribed it as “THE SEM IS IN E BIN KITCHEN A MOTHER IS WIPING DISHES”. Wav2Vec Standard outperformed the Manual Transcripts Unchanged methodology with these poorer transcripts. The reason for this improved performance with poorer transcripts is unclear and requires further examination of transcript quality and research. One possible explanation is that the AI transcripts were unable to capture “filler”/”function” words (pronouns, prepositions, conjunctions, and interjections) that do not convey as much meaning as “content” words (adjectives, nouns, verbs), which tend to be longer and more distinct.

### 4.6. Effect of Audio Enhancement

There was no clear advantage to enhancing the quality of audio files. In some cases, standard audio outperformed enhanced audio, while in others, enhanced audio performed better. Interestingly, using the more advanced cloud-based transcription programs, the standard audio performed consistently better. This worsened performance when using audio enhancement with cloud-based programs might be caused by the fact that these models have been trained with background noise in mind, and thus, the background noise removal of audio enhancement presents no advantages to these models, only disadvantages, as it might cause confusing noise artifacts. On the other hand, when using the Wav2Vec method (local), audio enhancement was extremely helpful, which indicates that it struggles heavily when presented with unclear audio. For Whisper, the other local method, there was no effect of using audio enhancement. Therefore, it would only make sense to use audio enhancement for locally-based ADAD systems.

### 4.7. The Most Effective Methodology for Real-World Applications

Since the real-world application of a speech-based automatic detection of AD program would be greatly affected by the distinction between using a locally-based and cloud-based methodology, it is of great importance to identify and differentiate between the best methodologies using each type of technology for future research and implementations. While a locally based ADAD service would have the advantage of perfect privacy and the lack of needing to pay for API or cloud fees (as all computations would be run locally and thus would not have to be saved on external servers), it would require the use of a powerful computer which could have high upfront costs. Alternatively, cloud-based systems need only minimal hardware (enough for a user interface) and a connection to the internet but would incur constant charges due to their use of cloud computing. Furthermore, a cloud system might cause privacy concerns among patients.

Based on the results of the AD and Control (230×) subgroup, the best methodology for a locally based system is the Wav2Vec Enhanced methodology. This methodology not only performed the best out of the locally based transcription methods, scoring 0.99 for both accuracy and F-1, but was the most effective method overall. The second best overall and best cloud-based methodology was the Talk Bank-proposed Rev AI methodology (using standard audio files). This methodology was able to score an impressive 0.96 for both accuracy and F-1 score. Overall, taking into account all ASR methods, neither system completely outperformed the other, showing that either type of implementation would be effective. Regardless, before any real-world implementation could be used, further research and testing would be necessary for either of these models.

### 4.8. Interpretation of Remaining Subgroups

The results of the AD, MCI, and Control (100×) and AD, MCI, Possible AD, and Control (100×) subgroups were promising but suffered heavily from overfitting, which is to be expected when using synthetic data. While SMOTE can be used to create synthetic data with a lower probability of suffering from overfitting, it is still possible. While the range of the AD, MCI, and Control (100×) using the train–test split was excellent, ranging from 0.98 to 0.90 for accuracy and 0.98 to 0.98 for F-1, the results of the CV test were quite poor. For accuracy, models only ranged from scoring 0.56 to 0.45, and for F-1, 0.55 to 0.43. While the train–test results are extremely promising, the CV results show that the models trained on this subset perform quite poorly when it comes to generalizability.

The most effective models for the AD, MCI, Possible AD, and Control (100×) subgroup using the train–test split were the IBM Cloud ones, which scored 0.96 (Enhanced) and 0.95 (Standard) for both accuracy and F-1. The worst model (Rev AI Enhanced) still did quite well, scoring 0.88 for accuracy and 0.87 for F-1 score. Similarly to the previous subgroup, the range for the train–test split test was very good, while the range of the CV scores was much poorer. The CV range for this data subgroup only spanned from 0.58 to 0.44 for accuracy and from 0.56 to 0.41 for F-1 score.

As discussed previously, a large discrepancy between the train–test and CV is highly indicative of overfitting. Since the Pitt Corpus is one of the largest databases of spontaneous speech, future research focused on collecting more data for MCI and Possible AD audio samples is necessary so that the results created by this study (for the final two subgroups) can be verified using original samples.

## 5. Conclusions and Future Research

The results of this research show that, with the current state of audio enhancement algorithms, AI-based ASR programs, AI-generated word embeddings, and machine learning classifiers, an accurate automatic speech-based AD detection system is possible. Furthermore, both these systems could be deployed through local or cloud-based computing, as both technologies produced machine-learning classification models that achieved near-perfect results when classifying between AD and a Control. To detect other diagnoses, such as MCI, more audio data are necessary for more accurate and reliable results.

Before any of these systems can be rolled out, more audio data (in addition to clinical trials) are necessary. These models were all trained using data from one specific area and time and, therefore, suffer from some intrinsic biases. A real-world application of this technology would need data from all over the world for each language, considering the various dialects and varying vernaculars that heavily influence speech. Unfortunately, current publicly available databases are highly limited, with the Pitt Corpus used by this study ranking as one of the largest databases available [4]. Therefore, the collection of new data and the creation of new databases are essential for the advancement of this technology.

Additionally, since data collection is vital to allow further research in this area, studies planning on creating new datasets should be sure not to include interviewers’ unaffected speech. This speech creates unhelpful biases in addition to being noisy data, lowering the overall effectiveness of the models produced.

Understanding why the poorer quality AI transcripts largely outperformed the higher quality manual transcripts is essential to further improving automatic AD detection. This would enable the further optimization of these proposed methodologies by enabling the removal of noisy data and giving insights into the parts of speech that are most important for speech-based detection systems.

## Figures and Tables

**Figure 1 brainsci-14-00211-f001:**
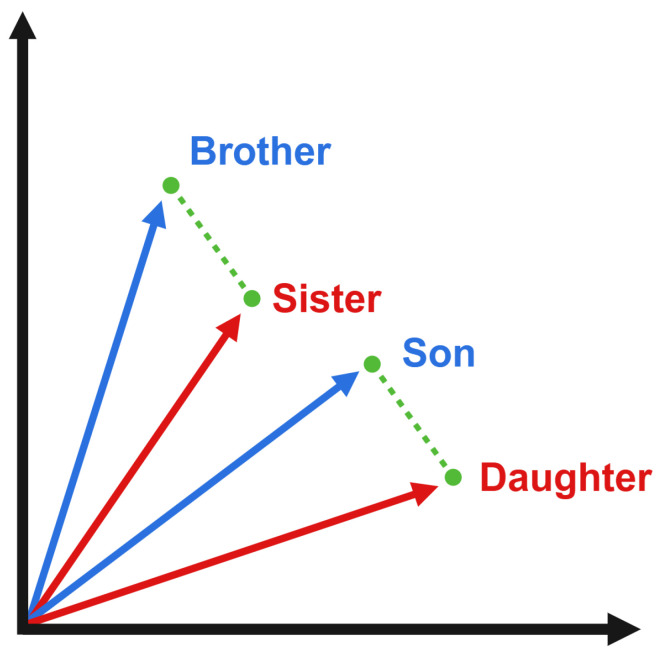
Two-degree vector graphical interpretation of N-degree vector word embeddings to convey linguistic meaning in a numerical format. The difference in meaning between “Brother” and “Sister” and “Son” and “Daughter” is identical and refers to the genders to which words in both groups of words apply; this equal difference can be seen through the identical vectors between them. Through these numerical interpretations of meanings, ML classifiers can be trained to detect patterns in text. Made with BioRender.

**Figure 2 brainsci-14-00211-f002:**
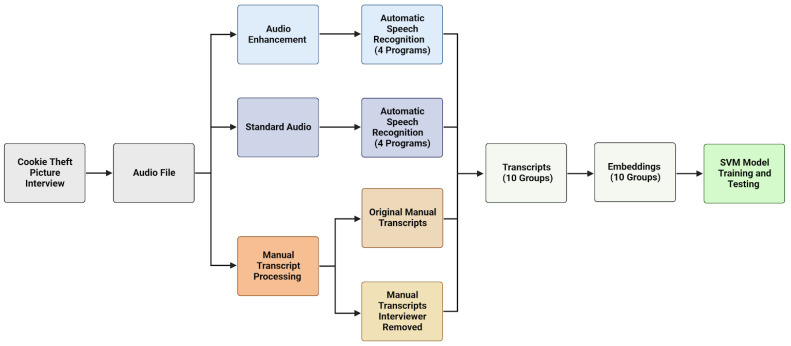
Overview of methodology for development and optimization of automatic spontaneous speech-based detection of Alzheimer’s disease. Audio files of patients describing an image (the Cookie Theft picture) were collected from the Pitt Corpus dataset of the Dementia Bank Database. The files included an original unedited version, an enhanced version using an implementation of Boll Spectral Subtraction [17], as well as a transcript in the CHAT file format. Each audio file (standard and enhanced) was transcribed using four different audio transcription services, while two original transcripts were generated, one from the original file and the other with the interviewer’s comments removed, creating a total of 10 transcription groups with different methodologies. These groups were turned into numerical representations using the second-generation OpenAI embedding model and were then used to train several SVM classification models. Made with BioRender.

**Figure 3 brainsci-14-00211-f003:**
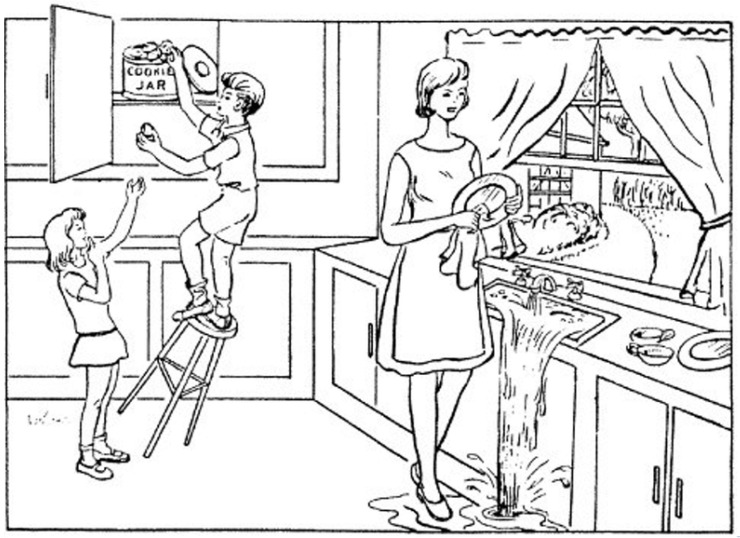
Cookie Theft picture from the Boston Diagnostic Aphasia Examination. This picture is shown to patients when conducting the Boston Diagnostic Aphasia Examination. Patients are asked to describe everything that they see in either a written or oral format. Patients’ descriptions are then used to identify issues with speech and fluency [18].

**Figure 4 brainsci-14-00211-f004:**
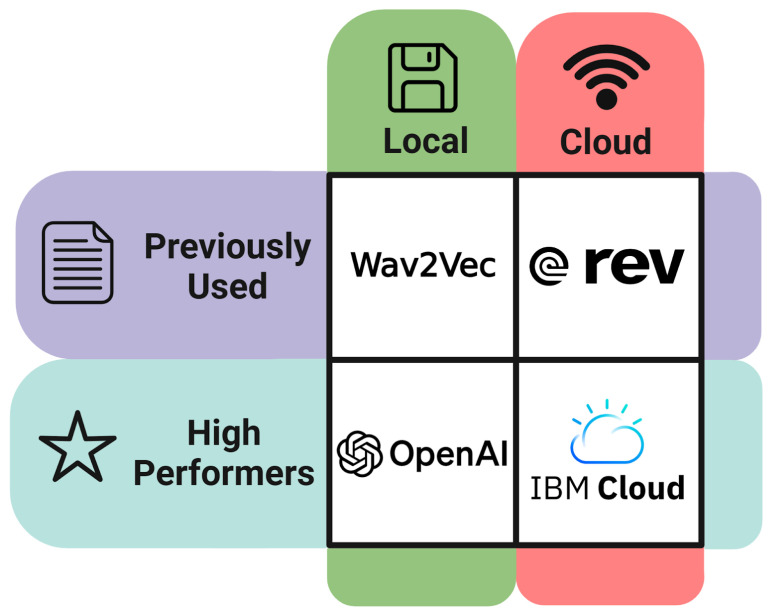
Automatic speech recognition using cloud-based and local system-based programs. Audio files were converted to text using four different ASR services. Two were cloud-based—IBM Cloud Watson and Rev AI (using the Talk Bank-developed interface)—and used a pay-as-you-go model as computations were performed remotely. OpenAI Whisper and Hugging Face Wav2Vec transcribed files locally using the computer’s own hardware and, accordingly, were free to use. At the same time, two of the ASR services (Wav2Vec and Rev AI) have been proposed in previous studies for this application, while the other two (OpenAI and IBM Cloud) have been shown to be industry leaders in transcription performance. Made with BioRender.

**Figure 5 brainsci-14-00211-f005:**
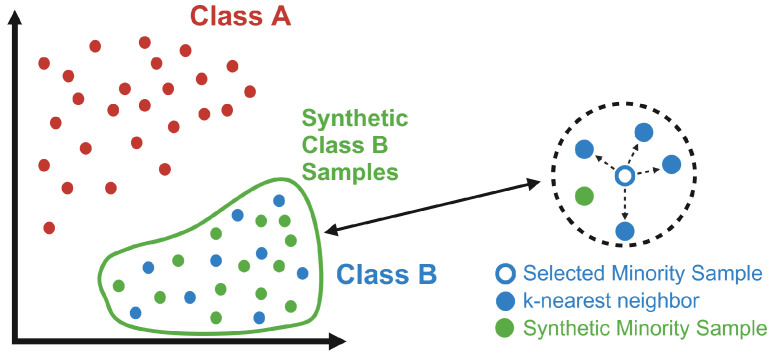
Visual interpretation of the Synthetic Minority Over-sampling Technique. Since there were large imbalances in the data used in this study, SMOTE was used to create synthetic data for the minority data classes. Specifically, the large difference between AD and Control, which both had around 230 samples, and MCI and Possible AD, which both had less than 50, meant that synthetic samples were needed for the latter classes in order to balance the dataset to produce effective classification models. As seen in the figure, SMOTE works by estimating the general area of the minority data groups (Class B) by selecting random minority data samples, calculating their distance from their k-nearest neighbors, and then generating synthetic samples with a similar distance from the selected data point. Made with BioRender.

**Figure 6 brainsci-14-00211-f006:**
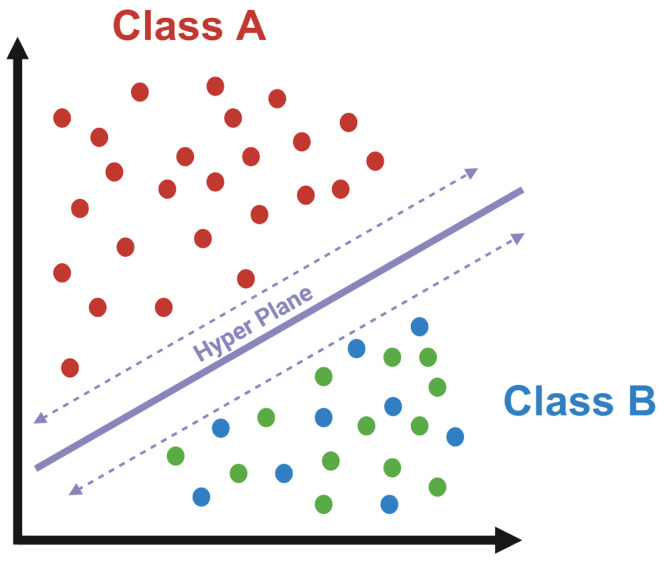
Visual interpretation of a support vector machine classifier. To classify data, an SVM classifier, or SVC, was used. An SVC was chosen to be the ML classifier based on past research, which indicated its increased performance when compared to other classifiers, such as Random Forest or neural networks. SVCs work by separating data groups with a hyperplane. This hyperplane is chosen in such a way that it maximizes the margin between the different classes of data. The data points closest to the hyperplane on either side are known as support vectors, and they essentially define the position and orientation of the hyperplane by acting as a margin. Made with BioRender.

**Table 1 brainsci-14-00211-t001:** Comparison of dataset size before and after processing and transcription.

Diagnosis	Probable AD	Possible AD	MCI	Control
Reported	244	87	23	121
Available	234	21	42	242
Transcribed	227	21	40	233

Only includes diagnosis types used in final models. Dataset versions, from top to bottom, refer to what was indicated by the database data sheet, what was available to download, and what was successfully transcribed by all methodologies.

**Table 2 brainsci-14-00211-t002:** Separation of transcript groups into four separate subgroups.

Data Group	Complete Transcribed Dataset	Transcribed Dataset Augmented with SMOTE	AD and Control (230×)	AD and Control (100×)	AD, MCI, and Control (100×)	AD, MCI, Possible AD, and Control (100×)
Probable AD	227	227	227	100	100	100
MCI	40	100	0	0	100	100
Possible AD	21	100	0	0	0	100
Control	233	233	233	100	100	100

All transcripts derived (manually and using ASR) from the Pitt Corpus of the Dementia Bank Database included samples from four diagnosis types: Control, AD, MCI, and Possible AD. As seen in the table, once minority classes were augmented using SMOTE, transcripts were organized into four subgroups. The names of each of these subgroups include the shortened name of each diagnosis contained as well as either (230×) or (100×) to indicate the number of samples for each diagnosis in the group. For data pools that were downsized for certain data groups, samples were randomly selected.

**Table 3 brainsci-14-00211-t003:** AD and Control (230×) results.

Transcript Group	Type	Model	Accuracy	Precision	Recall	F-1
Manual Transcripts	Unchanged	Train–Test Split	0.87	0.86	0.87	0.87
		10-Fold CV	0.82	0.82	0.82	0.82
	Participant Only	Train–Test Split	0.89	0.90	0.87	0.88
		10-Fold CV	0.84	0.85	0.84	0.84
IBM Cloud	Standard	Train–Test Split	0.94	0.94	0.94	0.94
		10-Fold CV	0.76	0.76	0.76	0.75
	Enhanced	Train–Test Split	0.91	0.90	0.91	0.91
		10-Fold CV	0.72	0.73	0.72	0.72
Rev AI	Standard	Train–Test Split	0.96	0.95	0.96	0.96
		10-Fold CV	0.77	0.78	0.76	0.76
	Enhanced	Train–Test Split	0.79	0.79	0.78	0.78
		10-Fold CV	0.77	0.77	0.76	0.76
Wav2Vec	Standard	Train–Test Split	0.88	0.88	0.87	0.88
		10-Fold CV	0.73	0.74	0.73	0.73
	Enhanced	Train–Test Split	0.99	0.99	0.99	0.99
		10-Fold CV	0.79	0.79	0.79	0.79
OpenAI Whisper	Standard	Train–Test Split	0.93	0.93	0.93	0.93
		10-Fold CV	0.81	0.81	0.81	0.81
	Enhanced	Train–Test Split	0.93	0.93	0.93	0.93
		10-Fold CV	0.80	0.81	0.80	0.80

Includes performance metrics for all 10 transcription types. Shows data for 80/20 train–test split model, which used 80% of data for training and 20% for testing, as well as the 10-Fold CV, which separated data into 10 folds and trained 10 models using a different fold for testing every time.

**Table 4 brainsci-14-00211-t004:** AD and Control (100×) results.

Transcript Group	Type	Model	Accuracy	Precision	Recall	F-1
Manual Transcripts	Unchanged	Train–Test Split	0.93	0.93	0.93	0.93
		10-Fold CV	0.74	0.77	0.74	0.73
	Participant Only	Train–Test Split	0.89	0.92	0.86	0.88
		10-Fold CV	0.80	0.85	0.80	0.79
IBM Cloud	Standard	Train–Test Split	0.98	0.97	0.98	0.98
		10-Fold CV	0.72	0.74	0.72	0.71
	Enhanced	Train–Test Split	0.89	0.88	0.89	0.88
		10-Fold CV	0.72	0.73	0.71	0.71
Rev AI	Standard	Train–Test Split	0.98	0.98	0.97	0.98
		10-Fold CV	0.63	0.66	0.62	0.60
	Enhanced	Train–Test Split	1.00	1.00	1.00	1.00
		10-Fold CV	0.65	0.67	0.65	0.64
Wav2Vec	Standard	Train–Test Split	0.84	0.83	0.84	0.84
		10-Fold CV	0.73	0.74	0.72	0.72
	Enhanced	Train–Test Split	1.00	1.00	1.00	1.00
		10-Fold CV	0.75	0.76	0.75	0.75
OpenAI Whisper	Standard	Train–Test Split	0.91	0.92	0.90	0.90
		10-Fold CV	0.79	0.81	0.79	0.79
	Enhanced	Train–Test Split	0.91	0.92	0.90	0.90
		10-Fold CV	0.78	0.80	0.78	0.77

Includes performance metrics for all 10 transcription types. Shows data for 80/20 train–test split model, which used 80% of data for training and 20% for testing, as well as the 10-Fold CV, which separated data into 10 folds and trained 10 models using a different fold for testing every time.

**Table 5 brainsci-14-00211-t005:** AD, MCI, Possible AD, and Control (100×) results.

Transcript Group	Type	Model	Accuracy	Precision	Recall	F-1
Manual Transcripts	Unchanged	Train–Test Split	0.90	0.91	0.89	0.89
		10-Fold CV	0.50	0.53	0.50	0.47
	Participant Only	Train–Test Split	0.90	0.90	0.90	0.89
		10-Fold CV	0.56	0.60	0.56	0.54
IBM Cloud	Standard	Train–Test Split	0.95	0.95	0.95	0.95
		10-Fold CV	0.51	0.53	0.51	0.50
	Enhanced	Train–Test Split	0.96	0.96	0.96	0.96
		10-Fold CV	0.52	0.55	0.52	0.51
Rev AI	Standard	Train–Test Split	0.94	0.94	0.94	0.94
		10-Fold CV	0.44	0.46	0.44	0.41
	Enhanced	Train–Test Split	0.88	0.87	0.87	0.87
		10-Fold CV	0.52	0.54	0.52	0.50
Wav2Vec	Standard	Train–Test Split	0.89	0.90	0.88	0.88
		10-Fold CV	0.58	0.60	0.58	0.56
	Enhanced	Train–Test Split	0.93	0.93	0.92	0.92
		10-Fold CV	0.57	0.58	0.57	0.56
OpenAI Whisper	Standard	Train–Test Split	0.89	0.89	0.89	0.89
		10-Fold CV	0.55	0.59	0.55	0.55
	Enhanced	Train–Test Split	0.94	0.93	0.94	0.93
		10-Fold CV	0.55	0.59	0.55	0.53

Includes performance metrics for all 10 transcription types. Shows data for 80/20 train–test split model, which used 80% of data for training and 20% for testing, as well as the 10-Fold CV, which separated data into 10 folds and trained 10 models using a different fold for testing every time.

**Table 6 brainsci-14-00211-t006:** AD, MCI, and Control (100×) Results.

Transcript Group	Type	Model	Accuracy	Precision	Recall	F-1
Manual Transcripts	Unchanged	Train–Test Split	0.97	0.97	0.96	0.97
		10-Fold CV	0.52	0.56	0.52	0.52
	Participant Only	Train–Test Split	0.98	0.99	0.98	0.98
		10-Fold CV	0.54	0.59	0.54	0.53
IBM Cloud	Standard	Train–Test Split	0.95	0.95	0.95	0.95
		10-Fold CV	0.53	0.54	0.53	0.52
	Enhanced	Train–Test Split	0.90	0.90	0.89	0.90
		10-Fold CV	0.55	0.55	0.55	0.54
Rev AI	Standard	Train–Test Split	0.91	0.92	0.91	0.90
		10-Fold CV	0.45	0.48	0.45	0.43
	Enhanced	Train–Test Split	0.94	0.93	0.94	0.93
		10-Fold CV	0.46	0.48	0.46	0.45
Wav2Vec	Standard	Train–Test Split	0.94	0.95	0.93	0.94
		10-Fold CV	0.55	0.59	0.55	0.52
	Enhanced	Train–Test Split	0.95	0.96	0.95	0.95
		10-Fold CV	0.55	0.55	0.55	0.53
OpenAI Whisper	Standard	Train–Test Split	0.92	0.93	0.91	0.92
		10-Fold CV	0.54	0.58	0.54	0.54
	Enhanced	Train–Test Split	0.94	0.94	0.93	0.93
		10-Fold CV	0.56	0.59	0.56	0.55

Includes performance metrics for all 10 transcription types. Shows data for 80/20 train–test split model, which used 80% of data for training and 20% for testing, as well as the 10-Fold CV, which separated data into 10 folds and trained 10 models using a different fold for testing every time.

## Data Availability

The data presented in this study are available on request from the corresponding author. The data are not publicly available due to privacy and ethical considerations surrounding the use of sensitive personal health information.

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
