# Peer review of "The Optimization of a Natural Language Processing Approach for the Automatic Detection of Alzheimer’s Disease Using GPT Embeddings"

_brainsci, 2024, doi:10.3390/brainsci14030211_

Round 1
Reviewer 1 Report
Comments and Suggestions for Authors
SMOTE is well known for its limitation when the number of dimensions of the feature space is high. The embedding size of text-embedding-ada-002 is about 1500, which is well above the range that SMOTE is generally considered to be effective (< a few hundreds). Moreover the embedding space for text classification is not continuous unlike image classification. The authors might want to investigate other text-specific data augmentation techniques.
Another issue is that it is not clear from the paper whether each split of the 10-fold CV is performed before or after data augmentation. Performing data augmentation before CV split could lead to data leakage between the train and test set, since two examples which originate from the same original datapoint may end up in such a way that one is in the train split and the other one is in the test split. Please confirm this issue and the result carefully.
Reviewer 2 Report
Comments and Suggestions for Authors
1. The title of the study includes the phrase "The Optimization of a Natural Language Processing Approach," but a classification has been performed using SVM with Embedding data. The contribution to the literature is not adequately clarified. The operation with SVM is a classification process, not an optimization. Therefore, the title does not accurately reflect the content of the study.
2. The novelity by using the SVM approach in the study has not been sufficiently highlighted in the text.
3. The results obtained with SVM should be compared with the results of other machine learning methods. The reason for choosing SVM exclusively has not been adequately stated. A comparison should be made, and the selection of SVM should be justified with results.
4. A flow diagram related to the study should be added.
5. The labels of the figures should be provided summarily. The sentences following the caption should be organized below the figure caption as a paragraph.
6. The use of active voice is present in some sections of the article, while passive voice is used in others. It is recommended to revise the entire article to consistently use either active or passive voice. Additionally, expressions like "My dataset was initially split..." are not in line with academic writing.
7. Tables have been included as figures. Tables should be added in table format.
Comments on the Quality of English LanguageThe use of active voice is present in some sections of the article, while passive voice is used in others. It is recommended to revise the entire article to consistently use either active or passive voice. Additionally, expressions like "My dataset was initially split..." are not in line with academic writing.
Round 2
Reviewer 1 Report
Comments and Suggestions for Authors
No further comment.
Reviewer 2 Report
Comments and Suggestions for Authors
Since the authors sent this study to a preprint service, it is seen that the ithenticate similarity rate is high, but no plagiarism is seen when looking at the similarities other than this similarity.
The corrections made by the authors after revision are sufficient for the article to be published in your journal.